# Associations between plasma nucleoside reverse transcriptase inhibitors concentrations and cognitive function in people with HIV

**Davide De Francesco**[1]*, **Xinzhu Wang**[2], **Laura Dickinson**[3], **Jonathan Underwood**[2,4], **Emmanouil Bagkeris**[1], **Daphne S. Babalis**[5], **Patrick W. G. Mallon**[6], **Frank A. Post**[7], **Jaime H. Vera**[8], **Memory Sachikonye**[9], **Ian Williams**[1], **Saye Khoo**[3], **Caroline A. Sabin**[1], **Alan Winston**[2], **Marta Boffito**[2,10], on behalf of the Pharmacokinetic and Clinical Observations in PeoPle Over fiftY (POPPY) study[¶]

1 Institute for Global Health, University College London, London, United Kingdom, 2 Department of Infectious Disease, Imperial College London, London, United Kingdom, 3 Department of Molecular & Clinical Pharmacology, University of Liverpool, Liverpool, United Kingdom, 4 Division of Infection and Immunity, University of Cardiff, Cardiff, United Kingdom, 5 Imperial Clinical Trials Unit, Imperial College London, London, United Kingdom, 6 Infectious Disease Epidemiology, University College Dublin School of Medicine, Dublin, Ireland, 7 King's College Hospital NHS Foundation Trust, London, United Kingdom, 8 Department of Global Health and Infection, Brighton and Sussex Medical School, Brighton, United Kingdom, 9 UK Community Advisory Board, London, United Kingdom, 10 Chelsea and Westminster Healthcare NHS Foundation Trust, London, United Kingdom

¶ Membership of the POPPY study group is provided in the Acknowledgments.
* d.defrancesco@ucl.ac.uk

**Data Availability Statement:** Within the limits of the ethical governance under which the data were collected, some restrictions have been placed on

## Abstract

### Objectives

To investigate the associations of plasma lamivudine (3TC), abacavir (ABC), emtricitabine (FTC) and tenofovir (TFV) concentrations with cognitive function in a cohort of treated people with HIV (PWH).

### Methods

Pharmacokinetics (PK) and cognitive function (Cogstate, six domains) data were obtained from PWH recruited in the POPPY study on either 3TC/ABC or FTC/tenofovir disoproxil fumarate (TDF)-containing regimens. Association between PK parameters ($AUC_{0-24}$: area under the concentration-time curve over 24 hours, $C_{max}$: maximum concentration and $C_{trough}$: trough concentration) and cognitive scores (standardized into z-scores) were evaluated using rank regression adjusting for potential confounders.

### Results

Median (IQR) global cognitive z-scores in the 83 PWH on 3TC/ABC and 471 PWH on FTC/TDF were 0.14 (-0.27, 0.38) and 0.09 (-0.28, 0.42), respectively. Higher 3TC $AUC_{0-24}$ and $C_{trough}$ were associated with better global z-scores [rho = 0.29 (p = 0.02) and 0.27 (p = 0.04), respectively], whereas higher 3TC $C_{max}$ was associated with poorer z-scores [rho = -0.31

this following discussion with members of the HIV community, reflecting the facts that the dataset contains very sensitive and potentially identifying information and that HIV is a highly stigmatised condition. Thus, our policy is to share data directly with interested parties for two purposes: 1) verification and replication of published analysis derived from the POPPY study, 2) novel scientific research projects using POPPY data. The POPPY steering committee includes a representative of the HIV community, Issues related to data collection and sharing have been discussed with members of the HIV community where the decision to impose data sharing restrictions has been made. The study was then approved by the UK National Research Ethics Service (NRES; Fulham London, UK number 12/LO/1409), based on the protocol including those restrictions. For data queries please contact Amalia Ndoutoumou, POPPY Trial Manager (poppy@imperial.ac.uk).

**Funding:** This work was supported by British HIV Association (BHIVA) research grant. The POPPY study is funded from investigator initiated grants from BMS, Gilead Sciences, Janssen, MSD and ViiV Healthcare (EudraCT Number: 2012-003581-40; Sponsor Protocol Number: CRO1992). The research is supported by the National Institute for Health Research (NIHR) Biomedical Research Centre based at Imperial College Healthcare NHS Trust and Imperial College London. The views expressed are those of the author(s) and not necessarily those of the NHS, the NIHR or the department of Health.

**Competing interests:** F.P. has received grants from Gilead, ViiV and Janssen and personal fees from Gilead, ViiV and MSD. C.S. has received funding from Gilead Sciences, ViiV Healthcare, and Janssen-Cilag for membership of data safety and monitoring boards, advisory boards, and speaker panels and preparation of educational materials. A. W. has received honoraria or research grants from ViiV Healthcare, Gilead Sciences, BMS, Merck and Co. and Janssen. M.B. has received speaker fees from Gilead, MSD/Merck and Janssen, advisory fees from ViiV, Gilead and MSD/Merck, honoraria from Gilead for speakers' bureau and a travel grant from Gilead, and has been the principal investigator in clinical trials sponsored by Gilead, ViiV, Mylan, Janssen and Bristol-Meyers Squibb. This does not alter our adherence to PLOS ONE policies on sharing data and materials.

(p<0.01)], independently of ABC concentrations. Associations of ABC PK parameters with global and domain z-scores were non-significant after adjustment for confounders and 3TC concentrations (all p's>0.05). None of the FTC and TFV PK parameters were associated with global or domain cognitive scores.

## Conclusions

Whilst we found no evidence of either detrimental or beneficial effects of ABC, FTC and TFV plasma exposure on cognitive function of PWH, higher plasma 3TC exposures were generally associated with better cognitive performance although higher peak concentrations were associated with poorer performance.

## Introduction

Whilst combination antiretroviral therapy (cART) has markedly improved the life expectancy of people with HIV (PWH), mild to moderate cognitive problems remain prevalent [1] and represent an important concern for PWH due to their potential impact on survival, quality-of-life and functional ability [2, 3].

Current cART regimens result in suppression of HIV RNA in both the plasma compartment and, in most individuals, the cerebrospinal fluid. Improvement in overall cognitive function is generally observed after commencing cART [4, 5]. However, cART-related neurotoxicity is often listed as a possible contributor to cognitive impairment in virally suppressed PWH [6]. Both direct and indirect mechanisms have been proposed to describe the role of cART toxicity in the development of cognitive problems in PWH, including interactions with vascular disease mechanisms and accelerated/accentuated brain ageing, induced mitochondrial dysfunctions, alterations of blood-brain barrier functionality, and direct peripheral nerve toxicity [7, 8]. In particular, the original nucleoside reverse transcriptase inhibitors (NRTIs) were associated with mitochondrial toxicities [9] and concerns exist whether modern NRTIs also exhibit similar toxicities which may affect the central nervous system (CNS).

Efficacy and neurotoxicity of cART regimens are likely to be driven by the concentration of each drug in both the plasma and CNS compartments as low concentrations may be associated with sub-optimally controlled HIV replication, whereas excessive concentrations may be associated with neurotoxicity. Nevertheless, limited clinical data exist on the effects, either beneficial or detrimental, of the exposure to different concentrations of specific cART agents, particularly NRTIs, on cognitive function of PWH. Here we investigated the association between cognitive function and plasma pharmacokinetics (PK) of four NRTIs that are currently in common use: lamivudine (3TC), abacavir (ABC), emtricitabine (FTC) and tenofovir (TFV), in a real-life cohort of PWH on cART.

## Methods

### Study design and participants

PWH were recruited in the POPPY study from HIV outpatient clinics in UK/Ireland from April 2013-January 2016 [10]. Inclusion criteria were: documented presence of HIV infection, white or black-African ethnicity, likely route of HIV acquisition via sexual exposure and ability to comprehend the study information leaflet. For the present analysis, we included those on cART regimens with an NRTI backbone of either 3TC/ABC or FTC/tenofovir disoproxil

fumarate (TDF), currently the two most commonly used NRTI combinations (tenofovir alafe-namide was not in widespread use in the UK at the time of study visit). Moreover, only participants who provided a plasma sample for PK testing and completed the assessment of cognitive function at the study visit were included. Socio-demographic and lifestyle characteristics were also collected via a structured interview with trained staff. The study was approved by the UK National Research Ethics Service (NRES; Fulham London, UK number 12/LO/1409). All participants provided written informed consent.

## Plasma NRTI concentrations and PK parameters

NRTI concentrations were measured by a validated method on ultra-performance liquid chromatography (ACQUITY, Waters) [11] in plasma samples collected at study baseline. Briefly, 200 µL of plasma was subjected to solid phase extraction (MCX cartridge, Waters) and the elutants were dried under nitrogen stream before being re-constituted in 50 µL of water. The NRTI drugs were separated using a C18 BEH column (1.7 µm, 2.1 mm x 100 mm, Waters). For each of the four NRTIs, population PK models were determined using nonlinear mixed effects (NONMEM v.7.3) using the $PRIOR subroutine to stabilise the models and aid partition of inter-individual and residual variabilities. The impact of covariates on antiretroviral apparent oral clearance was evaluated. Covariates included weight, age, sex, ethnicity, creatinine clearance (for TFV and FTC only) and the following genotypes: ABCC2 24C>T, ABCC2 1249G>A, ABCC10 526G>A, ABCC10 2843T>C (for TFV) and SCL47A1 G>A (for FTC). Univariate associations were assessed followed by multivariate analysis if more than one covariate was found to be significant. Model fit was assessed by statistical and graphical methods. A decrease in the minimal objective function value (-2*log likelihood) of at least 3.84 units (corresponding to a p-value<0.05) was required to accept a model with an extra parameter. Once significant covariates were incorporated, a backwards elimination process was performed and biologically plausible covariates generating an increase in the objective function value of at least 10.83 (i.e. p<0.001) units were retained. Using the final models, the following PK parameters were generated: area under the concentration-time curve from 0h to 24h ($AUC_{0-24}$), maximum concentration ($C_{max}$) and trough concentration ($C_{trough}$) as described previously [12, 13].

## Assessment of cognitive function

Assessment of cognitive function was performed using the computerized CogState battery, as reported previously [14]. The battery covered the six cognitive domains of visual learning, psychomotor function, visual attention, executive function, verbal learning and working memory. Raw test scores were log-transformed or arcsine root-transformed as recommended by the CogState guidelines for analysis and integrity and quality checks were applied to ensure that scores were generated from completed and fully understood tasks; individual test scores not meeting these checks were excluded from analysis. Individual test scores were converted into z-scores (mean of 0, standard deviation of 1) using the means and standard deviations obtained in the whole cohort of PWH. Domain z-scores were obtained by averaging individual test z-scores within the same domain, and a global z-score of overall cognitive function was obtained by averaging the six domain z-scores. For all z-scores, a higher value indicates better cognitive function.

## Statistical analysis

Analyses were conducted separately for those receiving 3TC/ABC and those receiving FTC/TDF. Continuous variables, including PK parameters and cognitive z-scores, were

summarized using the median and the interquartile range (IQR); categorical variables were described using frequencies and percentages. Associations between NRTI PK parameters and cognitive z-scores were assessed using a series of rank regression models, one for each combination of cognitive domain/global score, NRTI and PK parameter (7x4x3 models). In each model, the cognitive score was considered as the dependent variable and the NRTI PK parameter as an independent variable in addition to the following confounders, selected *a priori*: age, gender, ethnicity, education, recreational drug use, alcohol consumption, glomerular filtration rate (estimated using the Chronic Kidney Disease Epidemiology Collaboration equation [15]), depressive symptoms (assessed via the Patient Health Questionnaire-9 [16]) use of ritonavir/ cobicistat boosted protease inhibitor and use of efavirenz. In addition, for the analyses involving FTC and TFV PK parameters, body mass index (BMI) was added given it significantly correlated with FTC and TFV PK parameters. Multivariable analyses were conducted as follows, in order to investigate associations of each PK parameter for one NRTI, independently of the concentration of the other NRTI in the regimen. For each combination of cognitive domain/ global score, NRTI regimen (i.e. 3TC/ABC and FTC/TDF) and PK parameter, a rank regression model was performed (i.e. 7x2x3 models) with the simultaneous inclusion of the PK parameter of both NRTIs and the same confounders previously listed. All analyses were performed using the statistical software SAS v9.4 with p-values <0.05 considered as statistically significant.

## Results

### Characteristics of study participants

A total of 1073 PWH were recruited into the POPPY study, of which 1046 (97.5%) were on cART and 811 (75.6%) were on a cART regimen including either 3TC/ABC (n = 137) or FTC/ TDF (n = 674). Among these, 596 (73.5%) PWH provided a plasma sample of whom 554 (68.3%) completed the cognitive assessment and were therefore included in these analyses (n = 83 on 3TC/ABC, n = 471 on FTC/TDF). Overall, the majority of the 554 PWH were male (87.4%), of white ethnicity (89.5%) with a median (IQR) age of 52 (46, 58) years. The median (IQR) $CD4^+$ T-cell count was 660 (500, 850) cells/µL and 512 (92.6%) had a viral load <50 copies/mL (Table 1).

The median (IQR) global z-score was 0.11 (-0.28, 0.41) in all PWH, and 0.14 (-0.27, 0.38) and 0.09 (-0.28, 0.42) in those on 3TC/ABC and FTC/TDF, respectively (Table 1).

### Associations between PK parameters and cognitive z-scores

Univariable associations between PK parameters and cognitive scores, not accounting for potential confounders are shown in S1 and S2 Figs in S1 File. After adjusting for potential confounders, higher 3TC $AUC_{0-24}$ and $C_{trough}$ were associated with higher global z-scores [adjusted *rho* (95% CI) of 0.28 (0.02, 0.54) and 0.26 (0.01, 0.52), respectively, p = 0.03 and p = 0.05], with 3TC $C_{max}$ showing a negative association [adjusted *rho* (95% CI) = -0.28 (-0.51, -0.05), p = 0.02, Fig 1]. In particular, 3TC PK parameters were associated with the psychomotor, visual attention and working memory domains with similar patterns to those observed for the global z-score (S3 Fig in S1 File).

Associations of ABC PK parameters with global [adjusted rho (95% CI) of 0.13 (-0.10, 0.36), p = 0.27, 0.09 (-0.13, 0.32), p = 0.41 and 0.13 (-0.11, 0.37), p = 0.28 for $AUC_{0-24}$, $C_{max}$ and $C_{trough}$ respectively) and domain z-scores were weak and non-significant (Fig 1 and S3 Fig in S1 File). When 3TC and ABC PK parameters were evaluated simultaneously in the same model, higher 3TC $AUC_{0-24}$ and $C_{trough}$ were associated with greater global z-scores [adjusted *rho* (95% CI) of 0.26 (-0.01, 0.54), p = 0.06 and 0.24 (-0.03, 0.52), p = 0.09, respectively]

**Table 1. Characteristics of POPPY participants with PK and cognitive data.**

| n (%) or median (IQR) | All PWH (n = 554) | PWH on 3TC/ABC (n = 83) | PWH on FTC/TDF (n = 471) |
|---|---|---|---|
| Gender | | | |
| Male | 484 (87.4%) | 61 (73.5%) | 423 (89.8%) |
| Female | 70 (12.6%) | 22 (26.5%) | 48 (10.2%) |
| Age [years] | 52 (46, 58) | 52 (46, 59) | 52 (46, 58) |
| Ethnicity | | | |
| Black-African | 58 (10.5%) | 12 (14.5%) | 46 (9.8%) |
| White | 496 (89.5%) | 71 (85.5%) | 425 (90.2%) |
| Sexual orientation | | | |
| MSM/homosexual | 445 (80.3%) | 57 (68.7%) | 388 (82.4%) |
| Heterosexual | 109 (19.7%) | 26 (31.3%) | 83 (17.6%) |
| University degree or above | 255 (46.0%) | 45 (54.2%) | 210 (44.6%) |
| BMI [kg/m$^2$] | 25.5 (23.1, 28.0) | 25.2 (23.1, 27.8) | 25.5 (23.1, 28.0) |
| eGFR [mL/min/1.73m$^2$] | 92.1 (78.7, 102.0) | 93.9 (74.7, 105.8) | 90.7 (79.5, 101.3) |
| Recreational drug use | 168 (30.3%) | 21 (25.3%) | 147 (31.2%) |
| History of ID use | 66 (11.9%) | 11 (13.3%) | 55 (11.7%) |
| On boosted PI | 170 (30.7%) | 33 (39.8%) | 137 (29.1%) |
| On efavirenz | 184 (33.2%) | 15 (18.1%) | 169 (35.9%) |
| On 3TC/ABC | 83 (15.0%) | 83 (100.0%) | 0 (0.0%) |
| On FTC/TDF | 471 (85.0%) | 0 (0.0%) | 471 (100.0%) |
| Time since HIV diagnosis [years] | 12.8 (8.2, 19.3) | 15.5 (11.0, 20.3) | 12.3 (7.7, 18.9) |
| Nadir CD4$^+$ T cell count [cells/mm$^3$] | 216 (130, 310) | 205 (140, 304) | 218 (130, 310) |
| CD4$^+$ T cell count [cells/mm$^3$] | 660 (500, 850) | 700 (575, 865) | 656 (490, 831) |
| HIV RNA <50 copies/ml | 512 (92.6%) | 80 (96.4%) | 432 (91.9%) |
| AUC$_{0-24}$ [mg·h/l] | N/A | 3TC: 9.9 (7.7, 15.8) | FTC: 10.4 (9.2, 12.0) |
| | | ABC: 12.7 (11.0, 14.4) | TFV: 2.7 (2.4, 3.3) |
| C$_{max}$ [mg/l] | N/A | 3TC: 2.4 (2.3, 2.5) | FTC: 1.1 (1.1, 1.2) |
| | | ABC: 4.2 (4.0, 4.4) | TFV: 0.3 (0.2, 0.3) |
| C$_{trough}$ [μg/l] | N/A | 3TC: 12.4 (3.0, 126.7) | FTC: 75.0 (59.0, 99.0) |
| | | ABC: 2.6 (1.4, 4.6) | TFV: 53.0 (44.0, 67.0) |
| Time between last NRTI dose and PK sampling [hours] | | 14.2 (5.0, 17.0) | 12.9 (4.6, 16.0) |
| Global z-score | 0.11 (-0.28, 0.41) | 0.14 (-0.27, 0.38) | 0.09 (-0.28, 0.42) |
| Visual learning z-score | 0.10 (-0.39, 0.54) | 0.17 (-0.29, 0.51) | 0.09 (-0.44, 0.55) |
| Psychomotor z-score | 0.21 (-0.43, 0.62) | 0.13 (-0.50, 0.64) | 0.22 (-0.43, 0.62) |
| Visual attention z-score | 0.12 (-0.46, 0.59) | 0.18 (-0.43, 0.55) | 0.11 (-0.48, 0.60) |
| Executive function z-score | 0.20 (-0.38, 0.61) | 0.09 (-0.47, 0.57) | 0.21 (-0.38, 0.62) |
| Verbal learning z-score | 0.19 (-0.44, 0.77) | 0.23 (-0.17, 0.77) | 0.17 (-0.51, 0.75) |
| Working memory z-score | 0.12 (-0.34, 0.45) | 0.15 (-0.42, 0.50) | 0.11 (-0.34, 0.45) |

IQR: interquartile range; MSM: men having sex with men; BMI: body-mass index; ID: injection drug; PI: protease inhibitor.

whereas higher 3TC C$_{max}$ was associated with lower z-scores [adjusted *rho* (95% CI) of -0.27 (-0.50, -0.04), p = 0.02], with similar trends for the psychomotor, visual attention and working memory domains. On the other hand, ABC PK parameters did not show significant association with global and domain cognitive scores (all p's>0.05, Fig 1 and S3 Fig in S1 File).

Among PWH on FTC/TDF, none of the three PK parameters for either FTC or TFV was significantly associated with global or domain z-scores when these were considered individually and also when considered simultaneously (Fig 1 and S3 Fig in S1 File).

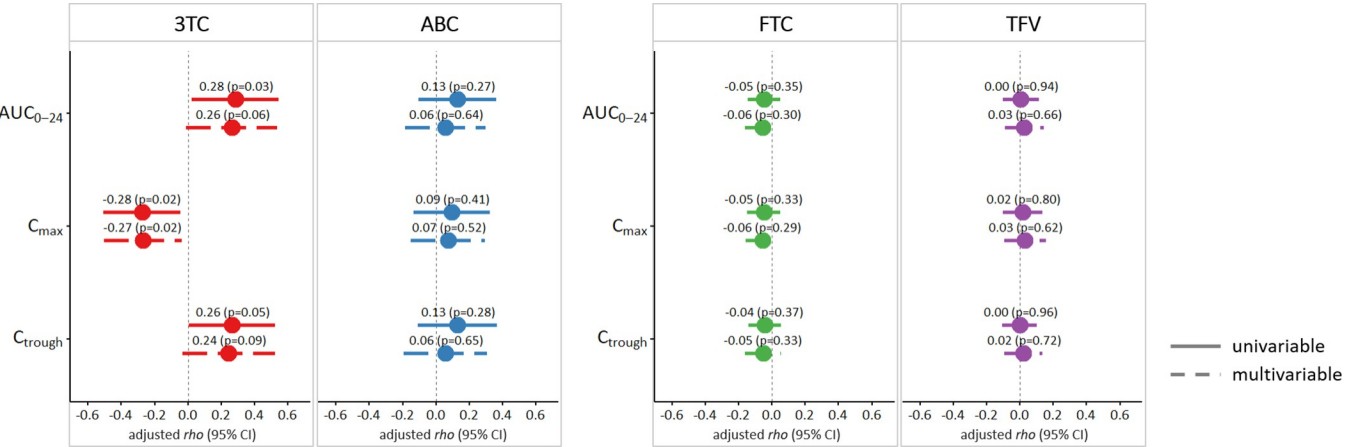

**Fig 1. Regression coefficients (i.e. adjusted rho) from rank regression evaluating the associations of PK parameters with global z-scores.** Associations are adjusted for age, gender, ethnicity, education, estimated glomerular filtration rate, use of ritonavir/cobicistat boosted protease inhibitor and use of efavirenz (3TC and ABC PK parameters), plus BMI (for FTC and TFV PK parameters only). Univariable estimates: one NRTI at the time in separate models; multivariable estimates: both NRTIs in the regimen in the same model (one model for each PK parameter).

## Discussion

We found no evidence of either detrimental or beneficial associations of ABC, FTC and TFV plasma exposure with cognitive function among PWH. In particular, controversies have existed with regards to the effect of ABC on cognitive performance, with one cohort study describing a beneficial effect [17] and a randomized controlled trial reporting no improvement of cognitive function after ABC initiation in PWH with HIV-associated dementia [18]. In our analysis, we did not observe any relationship between ABC concentration and cognitive scores.

Although high 3TC plasma exposures over the dosing interval of 24 hours and at the end of such interval were associated with greater cognitive performance, both overall and in some specific domains, maximum 3TC plasma concentrations were associated with poorer cognitive function. These findings may underline the importance of having sustained plasma drug exposure to ensure viral activity in all compartments. However, high 3TC peak concentrations may result in toxicity manifesting in poorer cognitive performance. If confirmed, this would suggest close monitoring of 3TC dosing and adherence to ensure optimal concentrations not exceeding thresholds potentially linked with toxicities. Importantly, we previously showed that higher 3TC exposure is associated with low glomerular filtration rates, suggesting that high 3TC $C_{max}$ may be observed in PWH with renal disease [12, 13]. Nevertheless, associations of 3TC PK parameters with cognitive scores were robust to potential confounding due to factors, including kidney function. We acknowledge that observing potential beneficial and potential detrimental effects with PK parameters and cognitive function with the one antiretroviral agent, here 3TC, within this one study, is not typical of what we may have expected to have observed, especially given the overall favourable safety profile of 3TC [19].

Our analysis has several limitations. Given our cross-sectional study, causal links cannot be assessed. Moreover, confounding due to adherence and other unmeasured factors may be additional sources of bias. Given the high number of statistical tests conducted, false positive findings may have occurred; therefore results should be interpreted with caution. Whilst we accounted for the concomitant use of efavirenz and boosted protease inhibitors, our sample size did not allow stratification according to individual third drugs in addition to the NRTI backbones considered. Therefore, we were not able to investigate how the potential beneficial or neurotoxic effect of a specific NRTI backbone may have changed depending on the third

drug. Importantly, we have assessed plasma PK parameters and not measurements of antiretroviral exposure intracellularly or in the CNS. Compared to plasma concentrations, CSF NRTI concentrations might be more directly associated with effects in the CNS, also based on the evidence that cART regimens with higher penetrations in the CNS are associated with lower HIV RNA levels and better cognitive function [20]. Cerebrospinal fluid and brain are two different compartments of the CNS and different concentrations of antiretroviral drugs can be found in the two compartments [21]. In particular, TDF has been shown to have low CNS penetration [22] and, consequently, plasma concentrations may not reflect CNS concentrations and related neurotoxic effects. Similarly, since NRTIs require intracellular activation, characterization of the intracellular level of the active triphosphate metabolite of NRTIs would have provided a better indication of virologic effectiveness and, therefore, of potential beneficial/detrimental effects on cognition than plasma NRTIs concentrations [23]. For these reasons, PK assessments in cerebrospinal fluid, brain tissue and other sites of the CNS, as well as characterizations of NRTIs intracellular metabolites, would be needed to provide some insight into antiretroviral drug exposure in the CNS and future work to assess exposure within this sanctuary site and any cognitive effects of such exposure are required. Finally, whilst regimens comprising a NRTI backbone remain the recognized standard of care, NRTI-sparing regimens have been investigated for their potential to reduce lifelong drug exposure and minimize the toxicity of NRTIs. Although limited, the evidence regarding potential difference between NRTI-based and NRTI-sparing regimens suggests no differences in cognitive function [24]. Nevertheless, further studies are needed to evaluate the effects related to concentrations of other classes of antiretroviral drugs such as protease and integrase inhibitors. Finally, at the time this study was conducted, tenofovir alafenamide was not in widespread use in the UK but is now one of the recommended NRTIs. Recent findings have shown no changes in cognitive performances in PWH switching from TDF to TAF, despite a significant reduction in TFV concentrations in CSF [25]. Therefore, it is unlikely that associations found would been different in more recent cohorts with higher rates of TAF use.

These limitations withstanding, these results could have implications for the development of treatment strategies for PWH with cognitive disorders, the identification of optimal drug concentrations and neurotoxicity thresholds, and for the design of future research programmes for PWH with or at risk of cognitive disorders.

## Supporting information

**S1 File.**
(DOC)

## Acknowledgments

We thank all participants and staff involved in the POPPY study. Membership to the POPPY study group (poppy@imperial.ac.uk) is as follows.

POPPY Management Team: Marta Boffito[1], Paddy Mallon[2], Frank Post[3], Caroline Sabin[4], Memory Sachikonye[5], Alan Winston[6, 7], Amalia Ndoutoumou[8], Daphne Babalis[8].

POPPY Scientific Steering Committee: Jane Anderson[9], David Asboe[1], Marta Boffito[1], Lucy Garvey[7], Paddy Mallon[2], Frank Post[3], Anton Pozniak[1], Caroline Sabin[4], Memory Sachikonye[5], Jaime Vera[10], Ian Williams[11], Alan Winston[6,7].

POPPY Sites and Trials Unit: Frank Post[3], Lucy Campbell[3], Selin Yurdakul[3], Sara Okumu[3], Louise Pollard[3], Beatriz Santana Suárez[3], Ian Williams[11], Damilola Otiko[11], Laura Phillips[11], Rosanna Laverick[11], Michelle Beynon[11], Anna-Lena Salz[11], Abigail Severn[11], Martin Fisher[10],

Amanda Clarke[10], Jaime Vera[10], Andrew Bexley[10], Celia Richardson[10], Sarah Kirk[10], Rebecca Gleig[10], Paddy Mallon[2], Alan Macken[2], Bijan Ghavani-Kia[2], Joanne Maher[2], Maria Byrne[2], Ailbhe Flaherty[2], Aoife McDermott[2], Jane Anderson[9], Sifiso Mguni[9], Rebecca Clark[9], Rhiannon Nevin-Dolan[9], Sambasivarao Pelluri[9], Margaret Johnson[12], Nnenna Ngwu[12], Nargis Hemat[12], Anne Carroll[12], Sabine Kinloch[12], Mike Youle[12] and Sara Madge[12], Amalia Ndoutoumou[8], Daphne Babalis[8], Alan Winston[6,7], Lucy Garvey[7], Jonathan Underwood[7], Lavender Tembo[7], Matthew Stott[7], Linda McDonald[7], Felix Dransfield[7], Marta Boffito[1], David Asboe[1], Anton Pozniak[1], Margherita Bracchi[1], Nicole Pagani[1], Maddalena Cerrone[1], Daniel Bradshaw[1], Francesca Ferretti[1], Chris Higgs[1], Elisha Seah[1], Stephen Fletcher[1], Michelle Anthonipillai[1], Ashley Moyes[1], Katie Deats[1], Irtiza Syed[1], Clive Matthews[1], Peter Fernando[1].

POPPY methodology/statistics/analysis: Caroline Sabin[4], Davide De Francesco[1], Emmanouil Bagkeris[1].

[1] St Stephen's Centre, Chelsea and Westminster Healthcare NHS Foundation Trust, London, UK

[2] HIV Molecular Research Group, University College Dublin School of Medicine, Dublin, Ireland

[3] Caldecot Centre, King's College Hospital NHS Foundation Trust, London, UK

[4] Institute for Global Health, University College London, London, UK

[5] UK Community Advisory Board, London, UK

[6] Department of Infectious Disease, Imperial College London, London, UK

[7] St. Mary's Hospital London, Imperial College Healthcare NHS Trust, London, UK

[8] Imperial Clinical Trials Unit, Imperial College London, London, UK

[9] Homerton Sexual Health Services, Homerton University Hospital, London, UK

[10] Elton John Centre, Brighton and Sussex University Hospital, Brighton, UK

[11] University College London Hospitals NHS Trust, London, UK

[12] Ian Charleson Day Centre, Royal Free Hospital NHS Trust, London, UK

## Author Contributions

**Conceptualization:** Patrick W. G. Mallon, Caroline A. Sabin, Alan Winston, Marta Boffito.

**Data curation:** Davide De Francesco, Jonathan Underwood, Emmanouil Bagkeris, Patrick W. G. Mallon, Frank A. Post, Jaime H. Vera, Ian Williams.

**Formal analysis:** Davide De Francesco, Xinzhu Wang.

**Funding acquisition:** Marta Boffito.

**Methodology:** Davide De Francesco, Xinzhu Wang, Laura Dickinson.

**Project administration:** Daphne S. Babalis, Memory Sachikonye.

**Supervision:** Saye Khoo, Caroline A. Sabin, Alan Winston, Marta Boffito.

**Visualization:** Davide De Francesco.

**Writing – original draft:** Davide De Francesco.

**Writing – review & editing:** Jonathan Underwood, Emmanouil Bagkeris, Patrick W. G. Mallon, Frank A. Post, Jaime H. Vera, Caroline A. Sabin, Alan Winston, Marta Boffito.

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
