## [Decision Letter · Decision Letter 0]

25 May 2021

PONE-D-21-12722

Associations between plasma nucleoside reverse transcriptase inhibitors concentrations and cognitive function in people with HIV

PLOS ONE

Dear Dr. De Francesco,

Thank you for submitting your manuscript to PLOS ONE. After careful consideration, we feel that it has merit but does not fully meet PLOS ONE’s publication criteria as it currently stands. Therefore, we invite you to submit a revised version of the manuscript that addresses the points raised during the review process.

We look forward to receiving your revised manuscript.

Kind regards,

Andrea Calcagno

Academic Editor

PLOS ONE

Additional Editor Comments:

This is an interesting study reporting plasma NRTI concentrations and neurocognitiove performance in PLWH from the POPPY cohort.

The reviewers commented on the lack of CSF concentrations and on the need for including TAF for novelty: despite I agree with them (although CSF/brain tissue PK correlation seems pretty poor), both are out of the Journal aims.

I believe the manuscript is valuable and it may improved through some of the changes they proposed (those that are feasable) and by acknowledging the missing data.

Journal Requirements:

"This work was supported by British HIV Association (BHIVA) research grant. The POPPY study is funded from investigator initiated grants from BMS, Gilead Sciences, Janssen, MSD and ViiV Healthcare (EudraCT Number: 2012-003581-40; Sponsor Protocol Number: CRO1992). The research is supported by the National Institute for Health Research (NIHR) Biomedical Research Centre based at Imperial College Healthcare NHS Trust and Imperial College London. The views expressed are those of the author(s) and not necessarily those of the NHS, the NIHR or the department of Health."

We note that you received funding from a commercial source: BMS, Gilead Sciences, Janssen, MSD and ViiV Healthcare

5. One of the noted authors is a group or consortium [Pharmacokinetic and Clinical Observations in PeoPle Over fiftY (POPPY) study]. In addition to naming the author group and listing the individual authors and affiliations within this group in the acknowledgments section of your manuscript and please also indicate clearly a lead author for this group along with a contact email address.

Reviewers' comments:

Reviewer's Responses to Questions

**Comments to the Author**

1. Is the manuscript technically sound, and do the data support the conclusions?

Reviewer #1: Yes

Reviewer #2: Partly

2. Has the statistical analysis been performed appropriately and rigorously? 

Reviewer #1: Yes

Reviewer #2: Yes

3. Have the authors made all data underlying the findings in their manuscript fully available?

Reviewer #1: Yes

Reviewer #2: Yes

4. Is the manuscript presented in an intelligible fashion and written in standard English?

Reviewer #1: Yes

Reviewer #2: Yes

5. Review Comments to the Author

Reviewer #1: PONE-D-21-12722 attempted to evaluate potential associations between NRTI concentrations and cognitive functions in PWH and showed no associations between certain NRTIs, but some with 3TC. In general, the MS was well written with sound background information, statistical analysis, results and discussion. However, the concentration analyses lack two key components in general: no CSF samples or active metabolites measured. The CSF samples, although remain a surrogate, might be more directly associated with CNS effects, e.g., cognitive function, using some newer references, such as "Switching to Tenofovir Alafenamide in Elvitegravir-Based Regimens: Pharmacokinetics and Antiviral Activity in Cerebrospinal Fluid". NRTIs are well known for their active metabolites and parent drug concentrations in plasma are poorly associated with biological effects. The authors should at least discuss these pitfalls more as part of limitations.

Minor:

The authors should also briefly discuss the NRTI sparing regimens and their potential impact on cognitive function in related to their findings presented in this MS.

Reviewer #2: To the Authors

1. Methods: “PWH were recruited in the POPPY study from HIV outpatient clinics in UK/Ireland from April 2013-January 2016”. What is happened thereafter? As one of the main limitation of the present study, no data from PWH on TAF were provided. The Authors acknowledged that “tenofovir alafenamide was not in widespread use in the UK at the time of study visit”. But, as matter of fact, 5 years have passed from the recruitment of patients. The inclusion of patients on TAF could add great value and novelty to the present investigation.

2. More information should be given on the methods used for the estimation of the AUC in the Methods section.

3. In the Discussion the Authors acknowledged, as potential study limitation, that they were not able to assess the concentrations of NRTIs in the CNS. This is, in my mind, a key information that cannot be missed. To strengthen the value of their findings, the Authors should provide data on CNS concentrations of NRTIs, at least in a cohort of patients, and try to correlate them with cognitive function.

4. As one of the main study findings, the Authors reported that higher plasma 3TC exposures were generally associated with better cognitive performance although higher peak concentrations were associated with poorer performance. In the Discussion the Authors should try to face with the clinical implications of these apparently inconsistent findings.

6. PLOS authors have the option to publish the peer review history of their article (what does this mean?). If published, this will include your full peer review and any attached files.

Reviewer #1: **Yes: **Qing Ma

Reviewer #2: No

---

## [Author Response · Author response to Decision Letter 0]

11 Jun 2021

Dear Prof. Andrea Calcagno,

We are grateful for the opportunity to revise our manuscript and value the constructive feedback that the editor and the reviewers have provided. Please see below our point-by-point responses to the reviewers’ comments along with changes made to the manuscript (page, paragraph and line numbers refer to the revised manuscript with tracked changes). 

Journal Requirements:

1. Please ensure that your manuscript meets PLOS ONE's style requirements, including those for file

naming. The PLOS ONE style templates can be found at https://journals.plos.org/plosone/s/file?id=wjVg/PLOSOne_formatting_sample_main_body.pdf and

RE: We amended our manuscript to meet PLOS ONE's style requirements

RE: We confirm that all information in the Funding Information is also present in the Financial Disclosure

"This work was supported by British HIV Association (BHIVA) research grant. The POPPY study is funded from investigator initiated grants from BMS, Gilead Sciences, Janssen, MSD and ViiV Healthcare (EudraCTNumber: 2012-003581-40; Sponsor Protocol Number: CRO1992). The research is supported by the National Institute for Health Research (NIHR) Biomedical Research Centre based at Imperial College Healthcare NHS Trust and Imperial College London. The views expressed are those of the author(s) and not necessarily those of the NHS, the NIHR or the department of Health."

We note that you received funding from a commercial source: BMS, Gilead Sciences, Janssen, MSD and ViiV Healthcare. Please provide an amended Competing Interests Statement that explicitly states this commercial funder, along with any other relevant declarations relating to employment, consultancy, patents, products in development, marketed products, etc.

Please know it is PLOS ONE policy for corresponding authors to declare, on behalf of all authors, all

potential competing interests for the purposes of transparency. PLOS defines a competing interest as anything that interferes with, or could reasonably be perceived as interfering with, the full and objective presentation, peer review, editorial decision-making, or publication of research or non-research articles submitted to one of the journals. Competing interests can be financial or non-financial, professional, or personal. Competing interests can arise in relationship to an organization or another person. Please follow this link to our website for more details on competing interests:

http://journals.plos.org/plosone/s/competing-interests

RE: Please see below our amended Competing Interests Statement.

“F.P. has received grants from Gilead, ViiV and Janssen and personal fees from Gilead, ViiV and MSD. C.S. has received funding from Gilead Sciences, ViiV Healthcare, and Janssen-Cilag for membership of data safety and monitoring boards, advisory boards, and speaker panels and preparation of educational materials. A.W. has received honoraria or research grants from VIiV Healthcare, Gilead Sciences, BMS, Merck and Co. and Janssen. M.B. has received speaker fees from Gilead, MSD/Merck and Janssen, advisory fees from ViiV, Gilead and MSD/Merck, honoraria from Gilead for speakers’ bureau and a travel grant from Gilead, and has been the principal investigator in clinical trials sponsored by Gilead, ViiV, Mylan, Janssen and Bristol-Meyers Squibb. The POPPY study is funded from investigator initiated grants from BMS, Gilead Sciences, Janssen, MSD and ViiV Healthcare. This does not alter our adherence to PLOS ONE policies on sharing data and materials.”

4. In your Data Availability statement, you have not specified where the minimal data set underlying the results described in your manuscript can be found. PLOS defines a study's minimal data set as the underlying data used to reach the conclusions drawn in the manuscript and any additional data required to replicate the reported study findings in their entirety. All PLOS journals require that the minimal data set be made fully available. For more information about our data policy, please see

http://journals.plos.org/plosone/s/data-availability. 

Important: If there are ethical or legal restrictions to sharing your data publicly, please explain these

restrictions in detail. Please see our guidelines for more information on what we consider unacceptable restrictions to publicly sharing data: http://journals.plos.org/plosone/s/data-availability#loc-unacceptabledata-access-restrictions. Note that it is not acceptable for the authors to be the sole named individuals responsible for ensuring data access.

RE: Please see below our amended Data Availability statement. 

“Within the limits of the ethical governance under which the data were collected, some restrictions have been placed on this following discussion with members of the HIV community, reflecting the facts that the dataset contains very sensitive and potentially identifying information and that HIV is a highly stigmatised condition. Thus, our policy is to share data directly with interested parties for two purposes: 1) verification and replication of published analysis derived from the POPPY study, 2) novel scientific research projects using POPPY data. To facilitate this, requests for data sharing can be made on a case-by-case basis following submission of a concept sheet to the POPPY principal investigators (c.sabin@ucl.ac.uk and alan.winston@imperial.ac.uk). Once submitted the proposed research/analysis will undergo review by the POPPY Steering Committee for evaluation of the scientific value, relevance to the study, design and feasibility, statistical power and overlap with existing projects. If the proposed analysis is for verification/replication, data will then be made available. If the proposed research is for novel science, upon completion of the review, feedback will be provided to the proposer(s). If the concept is approved for implementation, upon eventual revision, a writing group will be established consisting of the proposers and members of the POPPY study group.”

5. One of the noted authors is a group or consortium [Pharmacokinetic and Clinical Observations in PeoPle Over fiftY (POPPY) study]. In addition to naming the author group and listing the individual authors and affiliations within this group in the acknowledgments section of your manuscript and please also indicate clearly a lead author for this group along with a contact email address.

RE: As requested, we have amended the acknowledgments section to provide membership to the POPPY study group.

Review Comments to the Author:

Reviewer: 1

PONE-D-21-12722 attempted to evaluate potential associations between NRTI concentrations and cognitive functions in PWH and showed no associations between certain NRTIs, but some with 3TC. In general, the MS was well written with sound background information, statistical analysis, results and discussion. However, the concentration analyses lack two key components in general: no CSF samples or active metabolites measured. The CSF samples, although remain a surrogate, might be

more directly associated with CNS effects, e.g., cognitive function, using some newer references, such as "Switching to Tenofovir Alafenamide in Elvitegravir-Based Regimens: Pharmacokinetics and Antiviral Activity in Cerebrospinal Fluid". NRTIs are well known for their active metabolites and parent drug concentrations in plasma are poorly associated with biological effects. The authors should at least discuss these pitfalls more as part of limitations.

RE: We agree with the reviewer that CSF PK parameters would have provided insight of the penetration of NRTIs in the CNS and their potential neurotoxic effects. However, our study did not collect CSF samples or samples of other sites of the CNS to then assess PK parameters. Given the size of the cohort and the invasiveness of CSF sampling, this would have been unfeasible. We have expanded the discussion to elaborate more on this limitation and added a comment on the lack of NRTIs intracellular metabolites (page 10, 2nd paragraph, lines 10-14 and page 10/11, 2nd/1st paragraph, lines 18-19/1-3). 

Minor:

The authors should also briefly discuss the NRTI sparing regimens and their potential impact on cognitive function in related to their findings presented in this MS.

RE: As per the reviewer’s suggestion, we have added a paragraph discussing the potential relation between NRTI-sparing regimens and cognitive function (page 11, 1st paragraph, lines 5-10). 

Reviewer: 2

1. Methods: “PWH were recruited in the POPPY study from HIV outpatient clinics in UK/Ireland from April 2013-January 2016”. What is happened thereafter? As one of the main limitation of the present study, no data from PWH on TAF were provided. The Authors acknowledged that “tenofovir alafenamide was not in widespread use in the UK at the time of study visit”. But, as matter of fact, 5 years have passed from the recruitment of patients. The inclusion of patients on TAF could add great value and novelty to the present investigation.

RE: POPPY is a prospective study in which participants have been assessed at multiple time points. The analysis descripted in our manuscript refer to data collected at the 1st study visit, which took place between April 2013 and January 2016. At the time this analysis was conducted PK analysis of the plasma samples collected at a two-year follow up was still ongoing. We agree with the reviewer that these more recent data would add great value and clinical relevance, however this is out of the scope of the current study. We are planning to conduct a thorough longitudinal analysis of PK data and cognitive function that would complement the findings presented in our manuscript. For the current manuscript we limited to acknowledge this limitation while also referencing to recent findings showing no changes in cognitive performances when switching from TDF to TAF, despite a reduction in TFV concentrations in CSF (page 11, 1st paragraph, lines 11-15).

2. More information should be given on the methods used for the estimation of the AUC in the Methods section.

RE: As per the reviewer’s suggestion, further information regarding the estimation of PK parameters has been added in the methods section (page 5/6, 2nd/1st paragraph, lines 6-9/1-8).

3. In the Discussion the Authors acknowledged, as potential study limitation, that they were not able to assess the concentrations of NRTIs in the CNS. This is, in my mind, a key information that cannot be missed. To strengthen the value of their findings, the Authors should provide data on CNS concentrations of NRTIs, at least in a cohort of patients, and try to correlate them with cognitive function.

RE: As stated in the response to reviewer 1, CSF samples or samples of other sites of the CNS were not collected due to the large size of the cohort and the invasiveness of CSF sampling. Therefore we are unable to provide data on NRTIs concentrations in the CNS. Nevertheless, we have further commented on this issue in the discussion (page 10, 2nd paragraph, lines 10-14 and page 10/11, 2nd/1st paragraph, lines 18-19/1-3).

4. As one of the main study findings, the Authors reported that higher plasma 3TC exposures were

generally associated with better cognitive performance although higher peak concentrations were

associated with poorer performance. In the Discussion the Authors should try to face with the clinical implications of these apparently inconsistent findings.

RE: As per the reviewer’s suggestion, we have now added a sentence in the discussion highlighting the clinical implications of this finding (page 9, 4th paragraph, lines 6-7).

---

## [Editor Report · Decision Letter 1]

15 Jun 2021

Associations between plasma nucleoside reverse transcriptase inhibitors concentrations and cognitive function in people with HIV

PONE-D-21-12722R1

Dear Dr. De Francesco,

We’re pleased to inform you that your manuscript has been judged scientifically suitable for publication and will be formally accepted for publication once it meets all outstanding technical requirements.

Kind regards,

Andrea Calcagno

Academic Editor

PLOS ONE

Additional Editor Comments (optional):

I think the authors addressed all reviewers' comments that they were able to (CSF samples were not available).

I believe the manuscript may now be accepted as it is.
---

## [Editor Report · Acceptance letter]

12 Jul 2021

PONE-D-21-12722R1 

Associations between plasma nucleoside reverse transcriptase inhibitors concentrations and cognitive function in people with HIV 

Dear Dr. De Francesco:

I'm pleased to inform you that your manuscript has been deemed suitable for publication in PLOS ONE. Congratulations! Your manuscript is now with our production department. 

Kind regards, 

on behalf of

Dr. Andrea Calcagno 

Academic Editor

PLOS ONE